



# A Statistical Analysis of Rogue Waves in the Southern North Sea

Ina Teutsch[1], Ralf Weisse[1], Jens Moeller[2], and Oliver Krueger[1]

[1]Helmholtz-Zentrum Geesthacht, Max-Planck-Str. 1, 21502 Geesthacht
[2]Federal Maritime and Hydrographic Agency, Bernhard-Nocht-Straße 78, 20359 Hamburg

**Correspondence:** Ina Teutsch (ina.teutsch@hzg.de)

**Abstract.** A new wave dataset from the southern North Sea covering the period 2011-2016 and composed of wave buoy and radar measurements sampling the sea surface height at frequencies between 1.28-4 Hz was quality controlled and scanned for the presence of rogue waves. Here rogue waves refer to waves whose height exceeds twice the significant wave height. Rogue wave frequencies were analysed, compared to Rayleigh and Forristall distributions, and spatial, seasonal and long-term variability was assessed. Rogue wave frequency appeared to be relatively constant over the course of the year and uncorrelated among the different measurement sites. While data from buoys basically correspond with expectations from the Forristall distribution, radar measurement showed some deviations in the upper tail pointing towards higher rogue wave frequencies. Number of data available in the upper tail is, however, still limited to allow a robust assessment. Some indications were found that the distribution of waves in samples with and without rogue waves were different in a statistical sense. However, differences were small and deemed not to be relevant as attempts to use them as a criterion for rogue wave detection were not successful in Monte Carlo experiments based on the available data.

## 1 Introduction

Waves that are exceptionally higher than expected for a given sea state are commonly referred to as rogue waves (Bitner-Gregersen and Gramstad, 2016). What exactly "expected" and "exceptionally" mean is a matter of definition which is not addressed consistently throughout the literature (e.g., Dysthe et al., 2008). A common approach is to define a rogue wave as a wave whose height exceeds twice the significant wave height of the surrounding seas. Here, significant wave height refers to the average height of the highest third of the waves in a record, and is intended to correspond to the height estimated by a "trained observer".

The above definition of a rogue wave is based on a criterion developed by Haver and Andersen (2000). As rogue waves are often associated with incidents and damages to ships and offshore platforms (Haver and Andersen, 2000), these authors were primarily interested in whether or not such waves represent rare realisations of typical distributions of waves in a sea state. Based on 20-minute wave samples, Haver (2000) called a wave a rogue wave when it represented an outlier in reference to the second-order model commonly used in engineering design processes. He concluded that *"... the ratio of wave height to significant wave height that is likely to be exceeded in 1 out of 100 cases [in a second-order process] is about 2.0".* (Haver, 2000).



Since the late 1990s there has been an increasing number of studies analysing observed rogue waves or studying potential mechanisms for rogue wave generation. Such studies comprise the description and analysis of measurements of individual rogue wave events (e.g., Skourup et al., 1996; Haver, 2004; Magnusson and Donelan, 2013) or the description of rogue wave statistics from longer records (e.g., Stansell, 2004; Baschek and Imai, 2011; Christou and Ewans, 2014). Several studies contain

attempts to identify potential physical mechanisms of rogue wave formation, such as second-order nonlinearities (Fedele et al., 2016), modulational instability (Benjamin, 1967) caused by non-linear wave focusing (Janssen, 2003), or the directionality of the wave spectrum (Onorato et al., 2002). Soares et al. (2003) analysed laser records from the Draupner and North Alwyn platforms in the North Sea and found that rogue waves in stormy conditions here showed higher skewness coefficients and a lower steepness than waves simulated from second-order theory. They concluded that rogue waves must result from higher

than second-order models. Based on an analysis of waves from two locations in the North Sea and the North Atlantic, Olagnon and v. Iseghem (2000) reported that in high sea states, extreme waves occurred more frequently in seas steeper than on average. From the analysis of a large dataset, mostly from radars and lasers in the North Sea complemented with some data from other regions, Christou and Ewans (2014) on the other hand concluded that rogue wave frequencies were not governed by steepness and other parameters describing the overall sea state. Based on analyses of laser altimeter data, Stansell (2004) described rogue

wave frequencies to be only weakly dependent on significant wave height, significant wave steepness and spectral bandwidth. Cattrell et al. (2018) emphasised that predictors for rogue wave probability can probably not be derived for an entire dataset, but argued that location-specific forecasts might be possible.

So far, there is still no generally accepted picture and the overarching question raised by Haver and Andersen (2000) on whether rogue waves can be considered as "rare realizations of a typical population" or as "typical realizations of a rare

population" is still being debated. To address this question, a definition of what is "typical" for a given sea state and/or location is needed. In deep water and under the assumption that the sea surface represents a stationary Gaussian process, wave heights can be shown to be Rayleigh distributed (Holthuijsen, 2007). The Rayleigh distribution represents a special form of a Weibull distribution

$$P(H > cH_s) = \exp\left(-\frac{c^\alpha}{\beta}\right) \tag{1}$$

with parameters $\alpha = 2$ and $\beta = 0.5$. Here $P$ denotes the probability that the height $H$ of an individual wave exceeds the significant wave height $H_s$ by a factor $c$. Forristall (1978) analysed the frequency of large waves from 116 hours with hurricane wind speeds in the Gulf of Mexico. He found that for these cases the Rayleigh distribution substantially overestimated the frequency of large wave heights. From his data and analyses, he estimated that a Weibull distribution with parameters $\alpha = 2.126$ and $\beta = 0.5263$ provided a better fit to the observed data. In the ocean wave literature, a Weibull distribution with these

parameters is commonly referred to as the Forristall distribution.

To address the question on whether or not rogue waves represent typical realisations of such distributions, several studies compared them with data from observations. For stormy seas, Waseda et al. (2011) found that radar measurements were in agreement with expectations from a Weibull distribution with parameters close to those found by Forristall. Data from different types of instruments and different kinds of sea states were found to be located in-between Gaussian and second-order statistics





(Christou and Ewans, 2014). Magnusson et al. (2003) found an agreement of the majority of their laser and buoy measurement data with Rayleigh and Weibull distributions, but reported deviations from the known distributions in the upper tail. They were, however, undetermined about the significance of those deviations. Similar deviations from the Forristall distribution were reported by Forristall (2005) when individual 30-minute wave records were analysed. When the records were combined, the data were again found to fit the Forristall distribution. These results suggest that larger samples including rogue waves 65 might be needed to derive robust results.

In the present study, we analyse new data that has not been available for analysis before. Compared to previous studies the data set is large, comprising six common years of nearly uninterrupted measurements from eleven radar stations and wave buoys located in the southern North Sea. From these data, observed wave heights were compared with Rayleigh and Forristall distributions and seasonality, trends, and spatial correlation were assessed. It was further tested on whether or not information 70 from the background field may be derived that points towards increased rogue wave probability for given sea states.

## 2 Data and methods

### 2.1 Data

Six common consecutive years of sea surface elevation data from 2011 to 2016 were available from eleven measurement stations in the southern North Sea (Figure 1). At the five stations represented by red circles, radar devices are installed, measuring 75 air-gap to the water surface with a frequency of 2 Hz or 4 Hz, respectively. The six blue boxes mark surface-following Datawell Directional Waverider buoys of type MkIII, measuring at a frequency of 1.28 Hz. The buoy stations are located in the German Bight, while the radar stations are situated in the southern part of the North Sea, off the Dutch Coast and towards Great Britain. Table 1 provides an overview of the positions of the measurement stations and the water depth at each position.



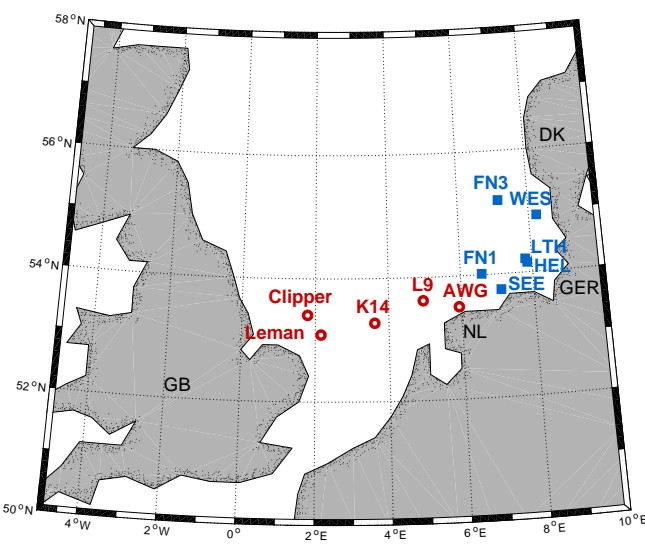

**Figure 1.** Wave measurement sites in the southern North Sea considered in this study. Blue squares: wave buoys, red circles: radar stations.

**Table 1.** Position and water depth at the measurement sites.

| Station name | Abbreviation | Latitude | Longitude | Water Depth |
|---|---|---|---|---|
| AWG | AWG | 53.493° | 5.940° | 6.3 m |
| L9 | L9 | 53.613° | 4.953° | 24 m |
| K14 | K14 | 53.269° | 3.626° | 26.5 m |
| Leman | Leman | 53.082° | 2.168° | 34 m |
| Clipper | Clipper | 53.458° | 1.730° | 21 m |
| Fino 3 | FN3 | 55.195° | 7.158° | 25 m |
| Westerland | WES | 54.917° | 8.222° | 13 m |
| Heligoland North | LTH | 54.219° | 7.818° | 30 m |
| Heligoland South | HEL | 54.160° | 7.868° | 20 m |
| Fino 1 | FN1 | 54.015° | 6.588° | 30 m |
| Norderney | SEE | 53.748° | 7.104° | 10 m |

The buoys delivered their data in the form of surface elevation samples, each of which had a length of 30 minutes (1800 s).
Radar data were available as continuous time series. For comparison, they were also split into half-hour samples. In total, the
procedure yielded approximately 797.000 half-hour samples from six years of observations at the eleven stations (Table 2).
Subsequently, all buoy and radar samples were treated equally.





In the following, a wave was defined as the course of the sea surface elevation in the time interval between two successive zero-upcrossings. Parameters describing the distribution of waves are found to be unaffected by the choice of upcrossing or downcrossing approaches (Goda, 1986). This way, a total of approximately 329 million individual waves were derived from the 797.000 samples.

**Table 2.** Number of available half-hour samples [$\times 10^4$] in 2011-2016 at each station after quality control (see Section 2.2). Measurement frequencies are indicated by font style: 1.28 Hz (normal text), **2 Hz (bold)**, *4 Hz (italic)*. The bottom row indicates data availability pear year in percent.

| Station/Year | 2011 | 2012 | 2013 | 2014 | 2015 | 2016 | total |
|---|---|---|---|---|---|---|---|
| **AWG** | **1.70** | *1.76* | *1.72* | *1.74* | *1.75* | *1.75* | 10.42 |
| **L9** | **0.96** | *1.46* | *1.75* | *1.75* | *1.75* | *1.75* | 9.42 |
| **K14** | **1.74** | *1.75* | *1.75* | *1.75* | *1.75* | *1.75* | 10.49 |
| **Leman** | **1.73** | *1.60* | *1.74* | *1.75* | *1.75* | *1.75* | 10.32 |
| **Clipper** | **1.69** | **1.70** | **1.60** | **1.70** | **1.71** | **1.71** | 10.11 |
| **FN3** | - | 0.76 | 1.21 | 1.07 | 1.51 | 1.16 | 5.71 |
| **WES** | - | 0.28 | 0.93 | 1.01 | 1.15 | 1.08 | 4.45 |
| **LTH** | 0.78 | 1.24 | 1.07 | 1.06 | 0.75 | 0.85 | 5.75 |
| **HEL** | - | 0.43 | 0.98 | 0.19 | - | 0.39 | 1.99 |
| **FN1** | 1.21 | 1.26 | 1.13 | 0.85 | 1.24 | 0.87 | 6.56 |
| **SEE** | 0.54 | 0.82 | 0.71 | 0.84 | 1.04 | 0.99 | 4.94 |
| **Data availability** | **54%** | **68%** | **76%** | **71%** | **75%** | **73%** | **69%** |

## 2.2 Quality control and rogue wave identification

Both, buoy and radar data were delivered in the form of raw surface elevation data. To identify and to eliminate spikes and erroneous data, each time series was checked and tested according to a number of quality criteria. These criteria were selected such that unreasonable spikes and data should be flagged and removed, while at the same time extreme peaks that may qualify as rogue waves should be maintained. In detail, the following procedure was applied to the raw samples:

1. Data within a 30-minute sample should be as complete as possible to allow for robust estimation of sea state parameters and individual waves. Samples missing more than three data points were discarded.

2. Since data were obtained not only during stormy but also in moderate and calm weather conditions, some samples contained a very large number of small waves. It was presumed that each wave in a record should be described by at least five measurement points to be reliably counted. When $n_p$ denotes the minimum number of measurement points per wave, the maximum number of waves $n_{max}$ in a 30-minute (1800 s) sample is given by $n_{max} = 1800\,\text{s}\, f_s\, n_p^{-1}$ where $f_s$




denotes the sampling frequency. For data from wave buoys sampled at a frequency of 1.28 Hz, thus 30-minute records

containing more than 460 waves were discarded. For the radar stations recording with sampling frequencies of 2 Hz and

4 Hz, samples containing more than 720 and 1440 waves respectively were excluded.

3. To eliminate influences from tides, the mean of each sample was subtracted. Subsequently, for each record, statistics such as significant wave height $H_s$, zero-upcrossing period $T_z$, and standard deviation $\sigma$ were calculated using the zero-upcrossing method. Significant wave height was computed as the average of the highest third of the waves in a 30-minute record.

4. Subsequently and based on physical reasoning a set of error indicators (EI) adopted from Christou and Ewans (2014) (EI 1-EI 5) and from Baschek and Imai (2011) (EI 6-EI 8) was applied. Time series were discarded if any of the error indicators was true:

**EI 1** A 30-minute sample included ten or more consecutive points of equal value.

**EI 2** A 30-minute sample included a wave with a zero-upcrossing period longer than $T_z = 25s$. For such waves to be

wind generated, extreme wind speeds exceeding hurricane strength over a fetch of more than 4.000 km for several hours would be required (WMO, 1998, p. 44), which appears unrealistic over the North Sea.

**EI 3** The limit rate of change $S_y$ of the water surface was exceeded. According to Christou and Ewans (2014) the limit rate is given by $S_y = 2\pi\sigma\overline{T_z^{-1}}\sqrt{2\ln N_z}$, where $\sigma$ represents the standard deviation of the surface elevation in the 30-minute sample and $\overline{T_z} = N(f_s N_z)^{-1}$ denotes the mean zero-upcrossing period. In the latter, $N$ denotes the number of

elevation points, $f_s$ again the sampling rate, and $N_z$ the number of zero-upcrossings in the sample. The criteria was applied for both, the surface elevation and its acceleration.

**EI 4** The energy in the wave spectrum at frequencies below 0.04 Hz (periods larger than 25 s) exceeded 5% of the total wave energy.

**EI 5** The energy in the wave spectrum at frequencies above 0.60 Hz exceeded 5%. These waves are too short for being

captured by five or more measurements at sampling frequencies of 1.28 Hz or 2 Hz.

**EI 6** The sample included at least one data spike for which the vertical velocity of the surface exceeded 6 ms$^{-1}$.

**EI 7** The ratio between the magnitudes of vertical and horizontal displacements exceeded a factor of 1.5 which, in deep water, is indicative of unexpected deviations from the orbital motions of the water particles.

**EI 8** At least one wave height in the sample exceeded the water depth.

5. The remaining samples were tested for the presence of rogue waves. They were considered to contain rogue waves if at least one of the waves in the sample fulfilled the criteria of Haver and Andersen (2000)

$$\frac{H}{H_s} \geq 2 \quad \text{and/or} \quad \frac{C}{H_s} \geq 1.25. \tag{2}$$

6. Detected rogue wave should again be described by at least five measurement points in order to be considered further.

7. Eventually all remaining rogues underwent a subjective visual check to ensure that all spurious extremes were removed.





Applying these criteria, in total approximately 28% of the buoy samples and 15% of the radar samples were eliminated and discarded from further analyses.

## 3   Results

Rogue waves refer to exceptionally high waves within a given sea state, where the state of the sea is commonly characterised by the significant wave height $H_s$. Whether or not a wave qualifies as a rogue under the definition of Haver and Andersen

(2000), thus does not directly depend on its height, but on its height relative to the height of the prevailing waves characterised by $H_s$. Rogue waves may hence occur in heavy seas but also during moderate or relatively calm conditions. Because the largest waves have largest impact, many studies focused on the analysis of extreme cases only; that is, the analysis of rogue waves for large $H_s$ (e.g., Forristall, 1978; Soares et al., 2003; Stansell, 2004; Waseda et al., 2011). Unlike these studies, in the following we use all available data from all sea states; that is, also cases with rogue waves from small or moderate sea states. In some

cases, when only rogue waves during high sea states are considered, this is explicitly mentioned. We generally analysed the number of rogue waves in relation to the total number of individual waves which in the following is referred to as rogue wave frequency.

### 3.1   Spatial distribution of rogue wave frequencies

Rogue wave frequency observed at the different stations within the period 2011-2016 varied between $1.24 \times 10^{-4}$ at WES

and $1.95 \times 10^{-4}$ at AWG (Figure 2). This corresponds on average to about 1.24 and 1.95 rogue waves in every 10.000 waves. Generally, rogue waves were detected more frequently in the radar than in the buoy samples. At all radar stations, rogue wave frequency exceeded the values expected from a Forristall distribution (Figure 2) while, with the the exception of SEE, values at buoy locations were below expectations from a Forristall distribution. Rogue wave frequencies are larger in the western part of our analysis domain but as all radar/buoy stations are located in western/eastern part of the domain we cannot infer whether this

is a result of the different measurement techniques or spatial location. When water depth is considered in addition (Table 1), no clear relation between rogue wave frequency and depth could be inferred.

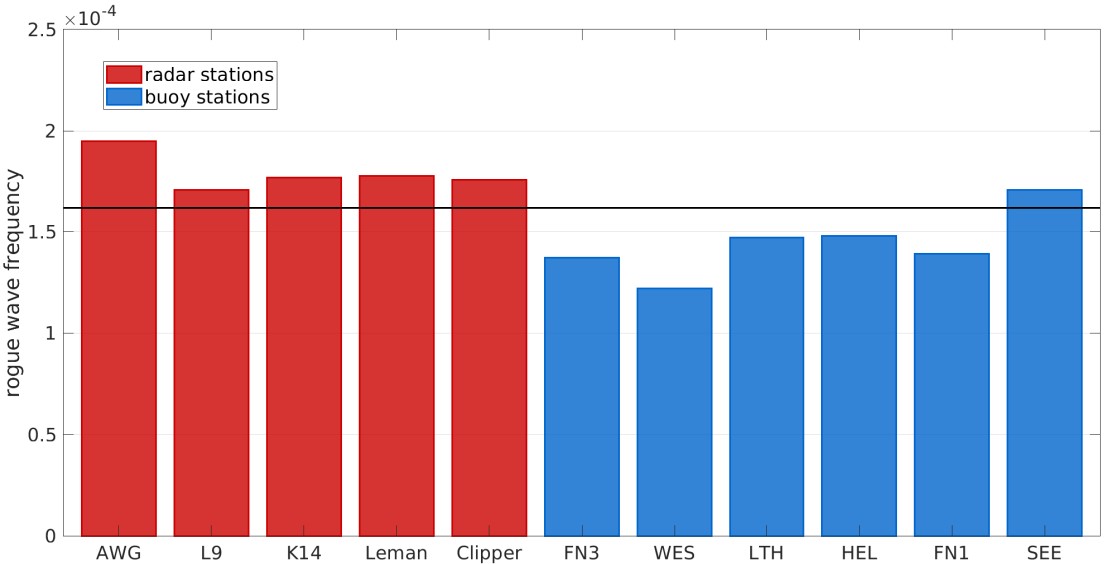

**Figure 2.** Rogue wave frequency in 2011-2016 at the eleven radar (red) and buoy (blue) locations. The solid black line indicates the rogue wave frequency ($1.62 \times 10^{-4}$) derived from the Forristall distribution (Forristall, 1978).

Spatial coherence between rogue wave frequencies at the different sites was analysed based on monthly values. Correlations were computed to test for the likelihood of joint occurrences of increased/decreased frequencies at the different stations for a given month. Only data from 2012-2016 were used because of larger gaps in 2011. Correlations between monthly rogue wave
frequencies at the different stations varied between -0.15 for K14 and HEL and +0.34 for Leman and FN1 (Table 3). For the given sample size of $N = 60$ monthly values these correlations are not significantly different from zero at the 95% confidence level. This indicates that monthly frequencies of rogue waves vary independently at the different stations.





**Table 3.** Correlations between monthly rogue wave frequencies 2012-2016 at the eleven measurement sites.

|  | AWG | L9 | K14 | Leman | Clipper | FN3 | WES | LTH | HEL | FN1 | SEE |
|---|---|---|---|---|---|---|---|---|---|---|---|
| **AWG** | +1.00 |  |  |  |  |  |  |  |  |  |  |
| **L9** | -0.01 | +1.00 |  |  |  |  |  |  |  |  |  |
| **K14** | +0.25 | +0.24 | +1.00 |  |  |  |  |  |  |  |  |
| **Leman** | +0.13 | +0.04 | +0.07 | +1.00 |  |  |  |  |  |  |  |
| **Clipper** | +0.04 | -0.06 | +0.03 | +0.17 | +1.00 |  |  |  |  |  |  |
| **FN3** | -0.06 | +0.11 | +0.01 | +0.05 | -0.12 | +1.00 |  |  |  |  |  |
| **WES** | -0.07 | -0.05 | -0.07 | +0.01 | -0.13 | +0.31 | +1.00 |  |  |  |  |
| **LTH** | -0.12 | +0.06 | +0.07 | +0.14 | -0.04 | +0.12 | -0.01 | +1.00 |  |  |  |
| **HEL** | -0.07 | +0.05 | -0.15 | -0.14 | -0.03 | +0.25 | +0.10 | +0.02 | +1.00 |  |  |
| **FN1** | -0.05 | -0.03 | +0.04 | +0.34 | +0.17 | +0.22 | +0.06 | -0.04 | -0.09 | +1.00 |  |
| **SEE** | -0.06 | +0.03 | -0.03 | +0.12 | +0.13 | -0.09 | -0.05 | +0.06 | +0.11 | -0.09 | +1.00 |

## 3.2 Temporal distribution of rogue wave frequencies

### 3.2.1 Seasonality

Rogue wave frequency; that is, the number of rogue waves per number of observed waves was found to be relatively constant and to vary only little in the course of the year (Figure 3). Even so a considerably higher number of rogue waves was observed during late summer and early fall. In absolute numbers, these waves are not necessarily high as significant wave heights in summer and early fall are generally small. In winter there a fewer rogue waves, but they generally occur during higher sea states and may thus have larger impacts. Moreover, wave periods are shorter in summer than in winter. Therefore, on average a

30-minute sample from the winter seasons contains fewer waves than a corresponding sample from summer (Figure 3). In total both effects cancel and rogue wave frequency was found to be remarkably stable in the course of the year. Similar conclusions hold, when the different measurement sites are analysed individually (Figure 4.)

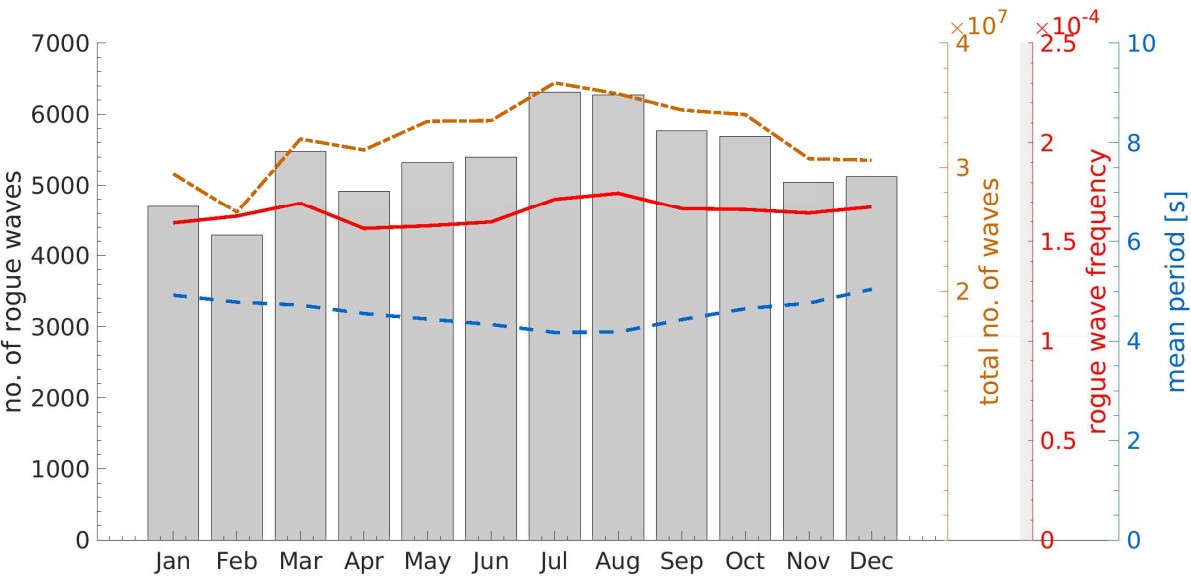

**Figure 3.** Seasonal distribution of rogue wave frequency (red), total number of waves (orange) and rogues waves (grey bars) in the period 2011-2016 and of monthly mean zero-upcrossing wave periods (blue) based on data from all measurement sites. Note the different scales and y-axis for the different parameters.

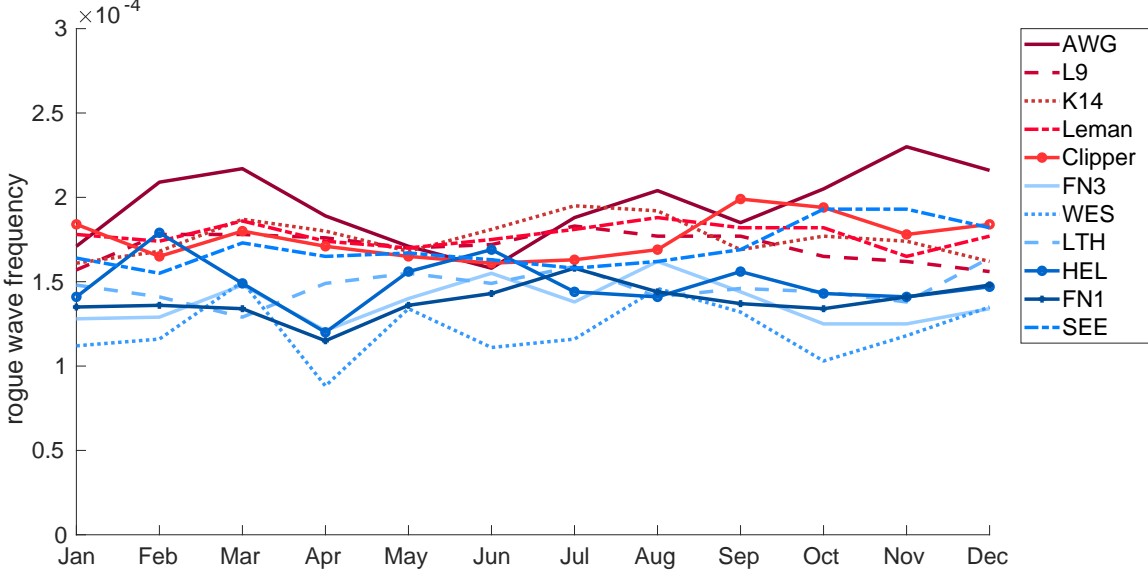

**Figure 4.** Seasonal distribution of rogue wave frequency in 2011-2016 at the eleven measurement sites (red colors: radar stations; blue colors: wave buoys).





### 3.2.2 Interannual variability

There was pronounced interannual variability in rogue wave frequency around its long-term mean at each measurement site
(Figures 5, 6). Variability was found to be somewhat larger at the radar stations in the western part of our domain. Largest
fluctuation where found at AWG where rogue wave frequency varied between -27% and 16.5% around the 2011-2016 mean.
Variability derived from the wave buoy data was somewhat smaller with the exception of the the two buoys WES and SEE,
both located in relatively shallow water (Table 1). Again, there is hardly any correlation between the values at the different
stations. While for example most stations suggest a minimum in rogue wave frequency for the year 2011, it was above average
at LTH. While LTH in turn showed very small frequencies in 2013, most other stations had values close to their long-term
means. Whereas AWG had a maximum in rogue wave frequency in 2014, other stations showed only small anomalies and
SEE even had low values in 2014. Albeit rogue wave frequency in 2016 was enhanced at most stations, this was not supported
by L9, Clipper and FN1. This further supports the results from the correlation analysis of monthly rogue wave frequencies
(Table 3). Despite the small distances between the measurement stations, rogue wave frequencies seem to vary independently.
This suggests that mechanisms driving rogue wave variability on larger scales might be difficult to identify.





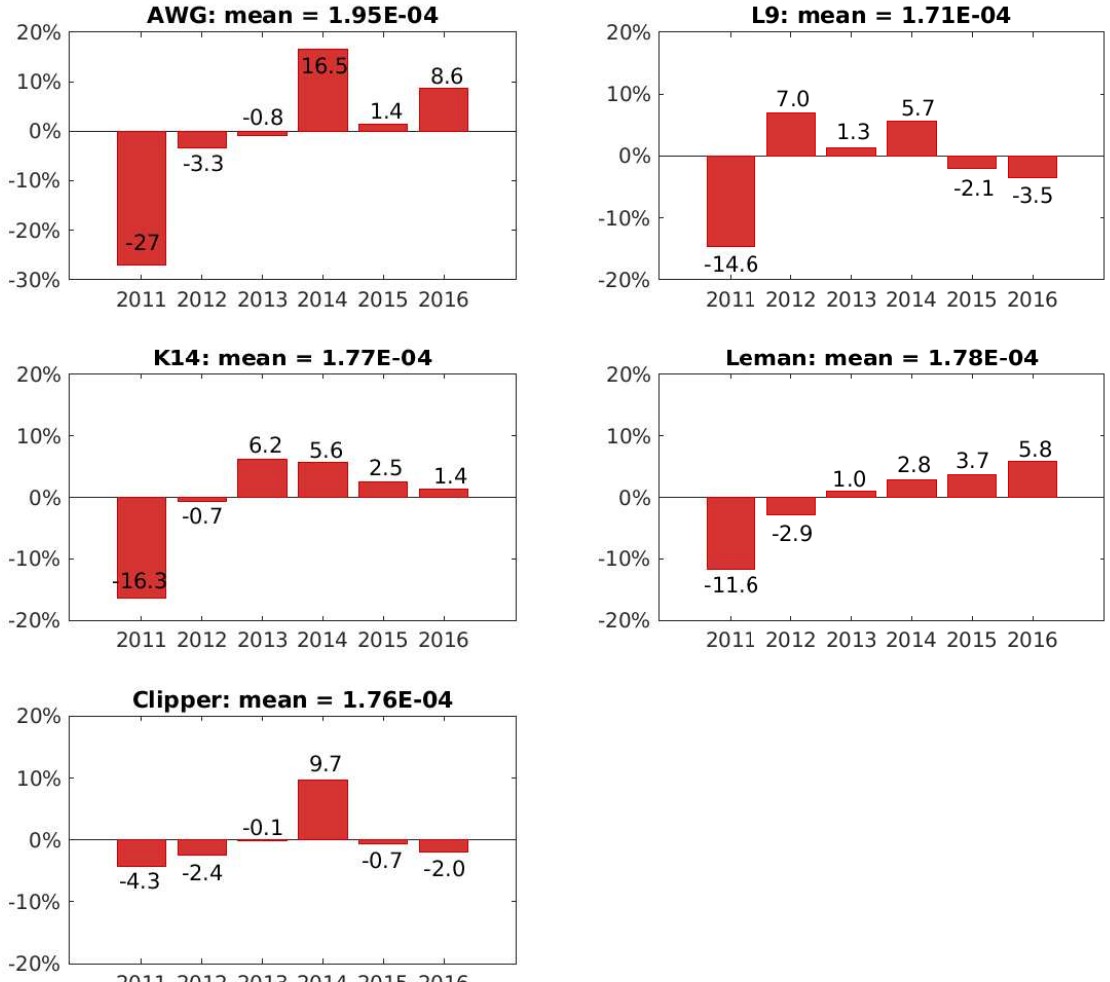

**Figure 5.** Anomalies in percent of annual rogue wave frequency relative to the corresponding long-term mean at each site for the five radar stations AWG, L9, K14, Leman and Clipper (from top left to bottom right).


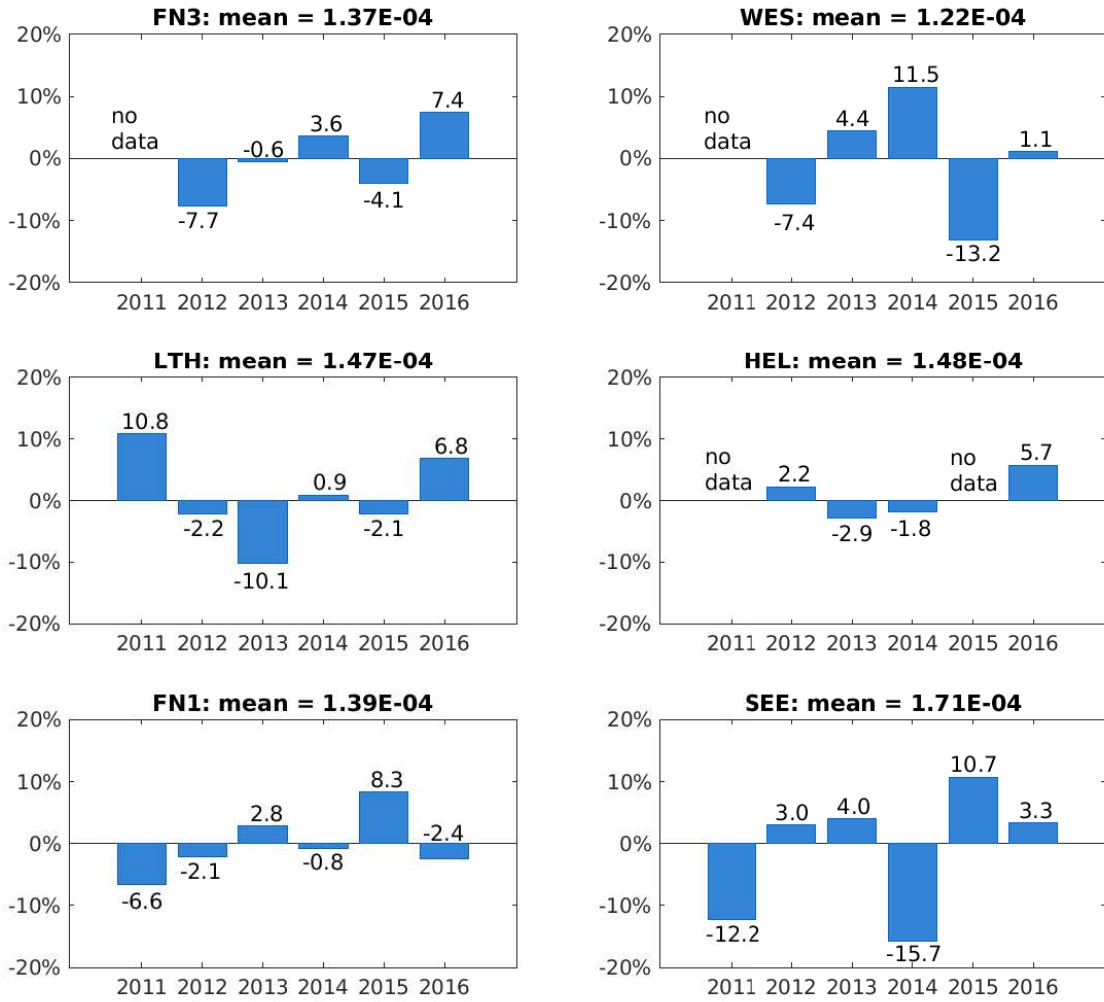

**Figure 6.** Anomalies in percent of annual rogue wave frequency relative to the corresponding long-term mean at each site for the six wave buoys FN3, WES, LTH, HEL, FN1 and SEE (from top left to bottom right).

### 3.3 Comparison of observations with Rayleigh and Forristall distributions

The cumulative frequencies of occurrences of wave heights relative to the significant wave height derived from the measurements were compared to corresponding exceedance probabilities given by Weibull distributions with both, Rayleigh and Forristall parameters (Figure 7). For wave heights up to twice the significant wave height, which corresponds to the threshold used to identify rogue waves, the measurement data are well described by the Forristall distribution. At a height of $H \approx 2H_s$

the data begin to deviate from the Forristall distribution. Both distributions increasingly diverge for larger relative wave heights $HH_s^{-1}$. This suggests that in our data rogue waves occurred more frequently than could be expected from the Forristall distribution. The frequency of rogue waves much larger than twice the significant wave height also exceeded expectations given by the Rayleigh distribution.

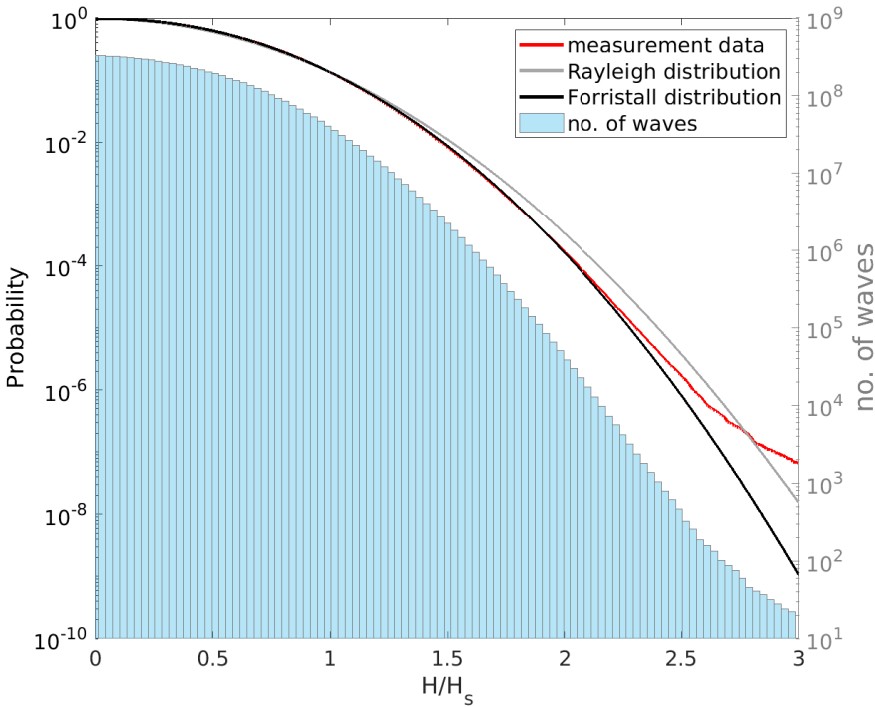

**Figure 7.** Comparison of cumulative distributions of relative wave heights from observations (red) to Rayleigh (grey) and Forristall (black) distributions together with number of observations in each relative wave height bin (blue bars). Note that the x-axis indicates relative wave height; that is, the height of each individual wave relative to the significant wave height of its 30-minute sample. The y-axis indicates the probability for relative wave heights to be exceeded.

To assess whether these deviations reflect a common behaviour or originate from a few measurement sites only, the analysis was repeated for each station individually (Figure 8). Substantial differences between the various sites were found. At AWG and Clipper, the frequency of waves higher than about twice the significant wave height increasingly deviated from the Forristall distribution and for waves larger than about 2.7 times the significant wave height reached or even exceeded that estimated from a Rayleigh distribution. This behaviour was found to be typical for the radar sites. On the other hand and with the exception of SEE, observations from the wave buoys generally followed (e.g., LTH) or underestimated (e.g. WES) frequencies from the Forristall distribution. Thus the radar stations were mostly responsible for the strong deviation of the overall dataset





from the Forristall distribution for extreme waves. This again may indicate differences arising from the different measurement techniques or the region.

So far the analyses were carried out for all sea states. For design purposes, navigation or other marine operations rogue
waves in high sea states that may causes largest damages are generally the most interesting ones. To assess whether a similar behavior is found also for these waves, the analysis was repeated including only cases in which the significant wave height exceeded the long-term 95%-tile at each site (Figure 9). Again a similar behavior as for all waves was found: For smaller waves, the frequency follows a Forristall distribution. The frequency of larger waves is substantially increased, in particular for rogue waves exceeding about 2.2 times the significant wave height. Again, the data from the radar stations accounted for most
of the deviation while data from the buoys followed the Forristall distribution more closely.

In summary, while results from the buoys (with the exception of SEE) suggest that rogue waves did not occur more frequently than could be expected from a Forristall distribution and thus could be considered as typical rare realisations within a given sea state, results from the radar measurements pointed towards enhanced rogue wave probability which might be indicative for mechanisms not described by second-order statistics. This holds for both, rogue waves in all and in high sea states only.

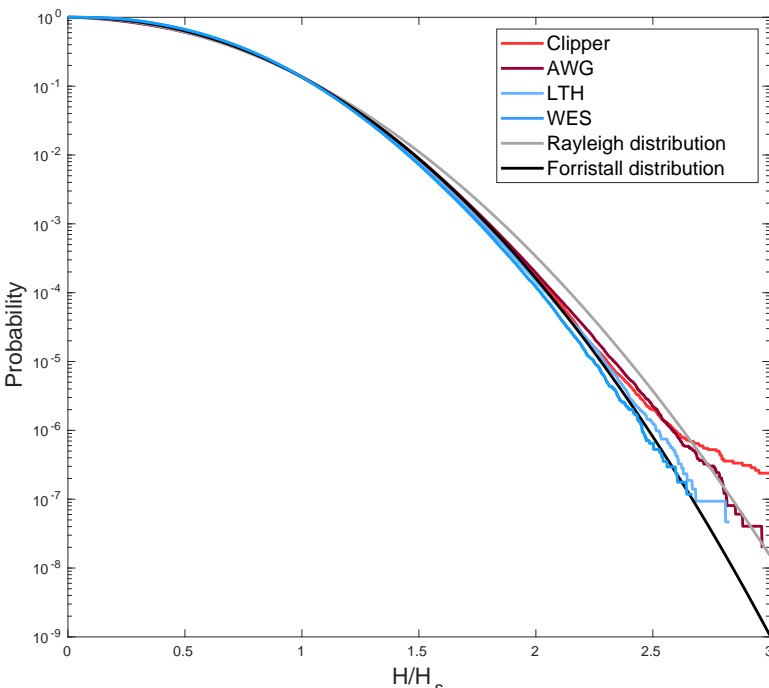

**Figure 8.** Comparison of the distributions of relative wave height at different stations to Rayleigh and Forristall distributions. On the x-axis, the height of each individual wave in relation to the significant wave height of its half-hour sample is given. The y-axis shows the probability for relative wave heights to be exceeded.

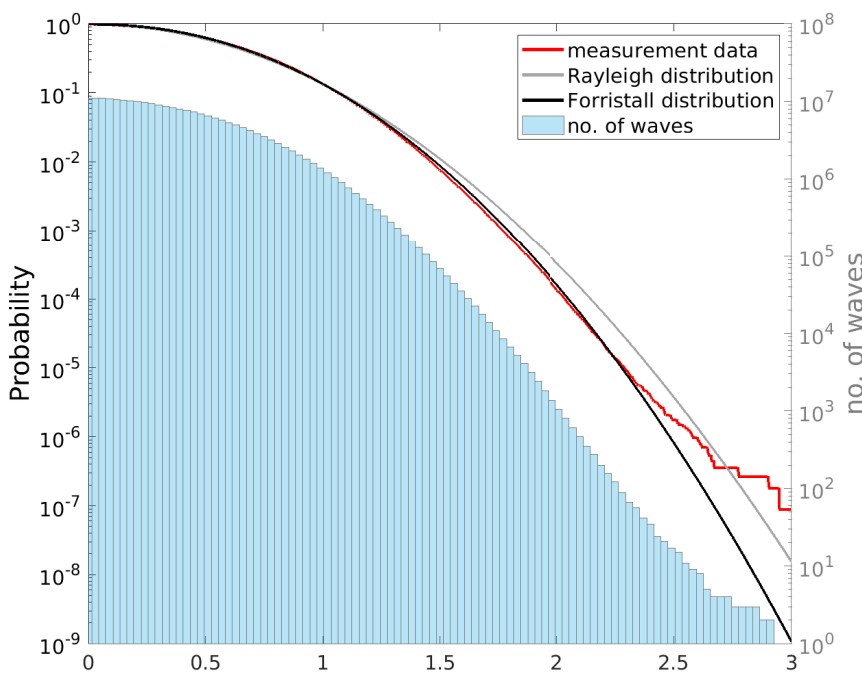

**Figure 9.** As Figure 7 but only including samples in which the significant wave height exceeded the corresponding long-term 95%-tile at the different sites.

## 3.4 Analysis of the background wave field

Data from some sites, especially the radar stations, suggested that differences between observed and the Forristall frequency distributions may exist for higher relative wave heights, in particular for those qualifying as rogue waves. In the following we distinguish between rogue waves and all other waves in 30-minute samples. The latter will be referred to as the background field. The aim was to investigate whether or not in samples with and without rogue waves differences in the distribution of waves in the background field might be identified that may bear a potential for rogue wave predictability.

More specifically, the measurement data were divided into two groups of samples: Group 1 comprised all samples including at least one rogue wave exceeding twice the significant wave height and Group 2 included all other samples. Subsequently, a third group was built from Group 1 by removing the individual rogue waves but retaining all other waves; that is the background field. In the following it is assessed to what extent differences in the background fields in Groups 2 and 3 can be identified.

### 3.4.1 Wave height distribution in the background field

The frequency distributions of wave heights in the background field in samples with and without rogue waves were compared (Figure 10). Visually, both distributions appear to be quite similar also the curve representing samples from Group 2 (normal


samples not containing rogue waves) is systematically below that of Group 3 (background field of samples containing rogue waves). This is supported by comparing the moments of the distributions with Group 2 having a slightly larger mean and

being marginally more flat-topped than Group 3 (Table 4). Additionally, the skewness of both distributions is positive, with the skewness of Group 3 being slightly more deviating from that of a normal distribution than Group 2.

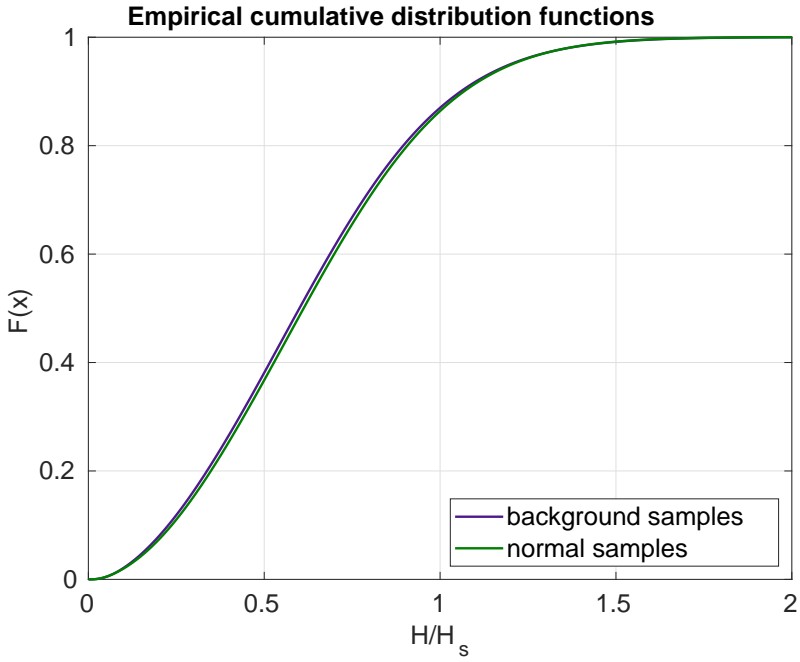

**Figure 10.** Empirical cumulative frequency distributions of relative wave heights in Groups 2 (green) and 3 (purple).

**Table 4.** Moments of the relative wave height distribution shown in Figure 10. Note that the relative wave height is non-dimensional.

|  | mean | standard deviation | kurtosis | skewness |
|---|---|---|---|---|
| **Group 2** | 0.638 | 0.319 | 2.944 | 0.473 |
| **Group 3** | 0.628 | 0.320 | 3.027 | 0.516 |

  To test whether the differences between the two groups were significant, a Kolmogorov Smirnov (KS) test (von Storch and Zwiers, 1999) was applied. More specifically, the KS test is a non-parametric test that compares two empirical distributions

and tests whether or not the null hypothesis that both distributions represent data from the same population can be rejected. The test is based on the distance $D$ between the two empirical distribution functions $F_{1,n}, F_{2,m}$ (Figure 10) such that

$$D_{n,m} = \sup_{x} |F_{1,n}(x) - F_{2,m}(x)| \tag{3}$$




where sup denotes the supremum function and $n, m$ the corresponding samples sizes. For large samples, the null hypothesis is rejected at level $\alpha$ when

$$D_{n,m} > K_\alpha \sqrt{\frac{n+m}{nm}}, \tag{4}$$


where

$$K_\alpha = \sqrt{0.5 \ln \frac{2}{\alpha}}. \tag{5}$$

For sample sizes of n = 306.282.148 waves in Group 2 and m = 23.073.717 waves in Group 3, the null hypothesis is to be rejected at $\alpha = 0.05$ when $D_{n,m} > 2.93 \times 10^{-4}$. From the data, $D_{n,m} = 1.42 \times 10^{-2}$ was estimated suggesting that the null

hypothesis that both samples originate from the same population should be rejected at at the 95% confidence level. This indicates that although differences appear to be small the test identified statistically significant differences between the background wave field from samples with and with rogue waves.

We suppose that this might be a consequence of the large sample sizes by which the test renders even very small differences as significant at a given significance level. We argue that for the differences to be *relevant*, they should further bear a potential

for rogue wave prediction or detection. To test this, a simple prediction/detection scheme was applied and tested for potential skill.

1. We split the data from Groups 2 and 3 into two halves and recomputed the cumulative distribution functions (cdf) of the first half.

2. From the second half, we 10.000 times randomly selected a 30-minute sample. In case the sample contained a rogue

wave, it was removed to only retain the background field. Subsequently, the empirical cdf of these data was computed.

3. Subsequently the distances between the empirical cdf and those of Group 2 and Group 3 (step 1) were computed respectively .

4. Based on the smaller of these distances, we predicted that a rogue wave was likely/unlikely to occur within the given sample.

5. We assessed whether or not the prediction would have been correct and marked the result correspondingly.

The results and the skill of this simple exercise are shown in Figure 11. It can be inferred that the probability of detecting a rogue wave correctly, given only the knowledge about the distribution of waves in the background field, is only about 55% ($POD = a(a+c)^{-1}$, Wilks (2011)). The probability of false detection $b(b+d)^{-1}$ (often referred to as false alarm rate, Barnes et al. (2009)) indicating how often a rogue wave would have been detected incorrectly, is about 41%. While this would still

imply some limited skill, the probability of a false alarm $b(a+b)^{-1}$ (often called the false alarm ratio, Barnes et al. (2009)) is extremely large and exceeds 90%. In total, the overall critical success index $a(a+b+c)^{-1}$ (Wilks, 2011), which refers to the number of correct yes forecasts divided by the total number of occasions on which the event was forecast and/or observed





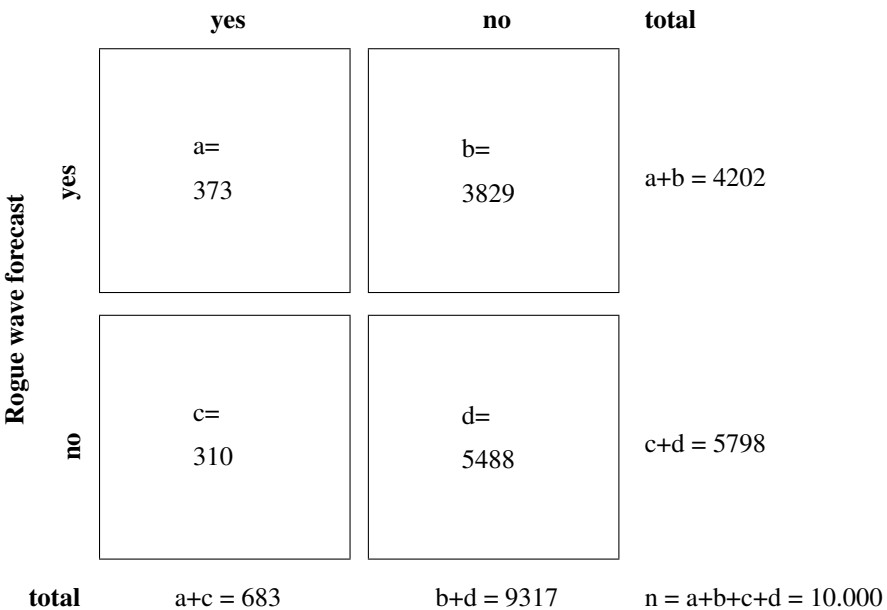

**Figure 11.** Contingency table of forecast/event pairs. a- hits. b- false alarms. c- misses. d- correct negatives.

is only about 0.08. For a perfect forecast, the critical success index would be unity. This suggests that although the KS test identified statistically significant differences between the distributions of wave heights in the background field of samples with

and without rogue waves, these differences appear not to be relevant as they hardly bear any potential for rogue wave detection or prediction. For an extended discussion about statistical significance and relevance, see e.g. Frost (2017). To test whether analyses done separately for the individual stations yield different results, the exercise was repeated only for stations that showed deviations from the Forristall distributions in the upper tail. In principle the same results were obtained. For example, the analysis of data from Clipper only yields a probability of detection of about 49%, a probability of false detection of about

46% and a probability of false alarm of 93%, very close to the values derived from the entire data set.

### 3.4.2 Wave steepness distribution in the background field

*Mean steepness*

Rogue waves are often described as exceptionally steep waves; that is, as waves whose height is large compared to their

length or period (Christou and Ewans, 2014; Donelan and Magnusson, 2017). In the following we investigate whether wave





steepness differs in samples with and without rogue waves. Following the approach taken in Christou and Ewans (2014), the mean wave steepness $S$ for each sample was derived from $S = H_s L^{-1}$ where $L$ denotes the mean wavelength in the sample. As both, wave buoys and radar devices provide point measurements, $L$ is not directly available but needs to be estimated from the wave period. A simplified approach was used to derive $L$ from the mean zero-upcrossing wave period $T_z$, averaged over

each 30-minute sample using linear wave theory. To distinguish between deep and shallow water waves, we used the critical water depth $kh = 1.36$, below which nonlinear instabilities are mostly absent (Benjamin and Feir, 1967). Here $k = 2\pi L^{-1}$ denotes the wavenumber and $h$ represents water depth. In principle, a distinction has to be made for each wave individually. For simplicity, in the following all waves from stations for which the shallow water condition was satisfied for the largest share of waves were treated as shallow water waves (AWG, SEE, WES) while waves at all other stations were considered as

deep water waves. For the shallow water stations, the mean wavelength $L$ of a sample was obtained from $L = \overline{T_z}\sqrt{gh}$ where $g$ and $h$ represent gravity acceleration and the water depth respectively. In deep water, the mean wavelength was computed by $L = (2\pi)^{-1}g\overline{T_z}^2$.

Following the approach taken by Christou and Ewans (2014), the maximum crest height in each sample was plotted as a function of mean wave steepness for both, samples with and without rogue waves (Figure 12). The analysis was performed

separately for shallow and deep water stations, as well as for radar and buoy stations. While the separation into deep and shallow water stations in principle led to different shapes of the scatter plot, in all cases rogue wave samples appear to be a subset of the samples without rogue waves. In other words, from the analysis it could not be inferred that the mean steepness in a rogue waves sample exceeds or systematically deviates from that of samples without rogue waves. This holds for both, wave buoys and radar stations. The same results as for the entire dataset were obtained when taking only stations into account that

showed deviations from the Forristall distribution in the upper tail.

**Figure 12.** Scatter plot between mean wave steepness and maximum crest height in samples with (red) and without (blue) rogue waves for shallow (top) and deep (bottom) water sites. Left: data from radar stations. Right: data from wave buoys.





*Steepness in the vicinity of a rogue wave*

While mean wave steepness was not found to systematically deviate between samples with or without rogue waves, such differences might still be limited to waves in the immediate vicinity of the rogue wave. Wilms (2018) investigated breaking waves in a hydrodynamic wave tank and observed increases in wave steepness five to six waves ahead of a breaking wave. To elaborate whether such a behaviour can also be found ahead of observed rogue waves in the real ocean, 1.234 rogue wave samples from radar devices and 716 rogue wave samples from wave buoys were used to derive a distribution of wave steepness of individual waves ahead of the rogue wave (Figure 13). Only severe sea states where considered; that is, only samples were regarded in which the significant wave height exceeds the corresponding long-term 95% percentile at each station. This was done as determining the shape and steepness of individual waves was more robust and reliable for high waves with large periods.

For both, radar and wave stations, the rogue waves themselves stick out as waves of strongly increased wave steepness in the order of about twice that of the preceding waves. The distributions of the 2-10 waves ahead of the rogue waves were not peculiar noticeable. All of them were characterised by almost constant median steepnesses ranging between about 0.037-0.041 at radar and between about 0.032-0.034 at wave buoy locations. Only the waves directly ahead of the rogue wave showed a tendency towards increased wave steepness (0.054 and 0.036 for radar and buoy stations respectively). However, the latter strongly depends on the choice of the method used to define the waves. In our analyses, a zero-upcrossing approach was used. In this case the trough preceding a rogue wave is considered to be part of the wave ahead. When zero-downcrossings would have been used instead, the wave trough preceding the rogue wave would have been treated as a part of the rogue wave itself. Since the wave trough ahead of a rogue wave is usually not as deep as the one following it, this would have led, in most cases, to a decrease in the steepness of the rogue and its preceding wave. Consequently, such a definition would have supported the conclusion that also the steepnes of the wave immediately ahead of the rogue wave is not outstanding compared to the others.



Natural Hazards
and Earth System
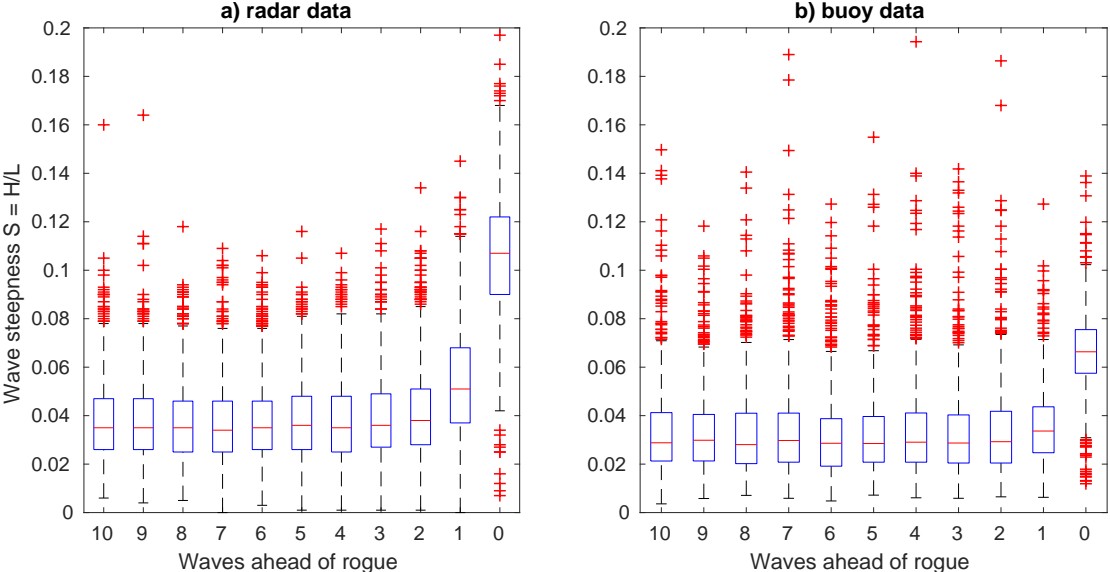

**Figure 13.** Distribution of wave steepness of the ten individual waves preceding a rogue wave (wave 0) for radar (left) and wave buoy (right) locations. Distributions were obtained from 1.234 (716) rogue wave samples at radar (buoy) locations for which the significant wave height exceeded the corresponding long-term 95% percentile. Distributions are shown as Box-Whisker plots (median: red line; box: inter-quartile range; whiskers: $1.5 \times$ inter-quartile range; red crosses: data outside the whiskers).

### 3.4.3 Asymmetry of waves preceding rogue waves

For steep waves, such as rogue waves, due to nonlinear wave-wave interactions higher wave crests are expected compared to
second-order theory (Forristall, 2005; Christou and Ewans, 2014). This results in asymmetric waves where the asymmetry $\mu$ can be described as the ratio between crest height $C$ and wave height $H$. For linear sine waves, the asymmetry is $\mu = 0.5$; for second-order Stokes waves in deep water, it is $\mu = 0.61$ (Wilms, 2018). The parameter $\mu$ is commonly used for the description of the geometry of breaking waves (Kjeldsen and Myrhaug, 1980). According to Kjeldsen and Myrhaug (1980), the asymmetry of breaking waves may reach values of up to $\mu = 0.84 - 0.95$. For rogue waves, Magnusson and Donelan (2000) stated that
they are characterised by pronounced crest-to-trough asymmetries, similarly to breaking waves. From wave tank experiments, Wilms (2018) concluded that increased asymmetries may occur five to six waves ahead of breaking waves.

Using the same rogue wave samples of 1.234 radar and 716 buoy data as above and in which the significant wave height exceeded the long-term 95%, distributions of wave asymmetries of the waves preceding the rogue waves were computed (Figure 14). Generally and on average, for both radar and wave buoy stations, asymmetries of the 2-10 waves preceding the rogue
wave were close to the value of $\mu = 0.5$ expected from linear theory. The waves immediately ahead of the rogue waves, on average showed a strong decrease in asymmetry, while asymmetry of the rogue waves themselves was increased, indicating higher crests than troughs. Again, this result strongly depends on how the individual waves were defined. The reduced asymmetry of the wave immediately ahead of the rogue wave is due to the assignment of the relatively deep trough ahead of the





rogue to the preceding wave. Using a zero-downcrossing analysis, this trough is assigned to the rogue wave and the mean

asymmetry remains constant at approximately 0.5 with the exception of the rogue wave itself. Additionally, it is interesting to note that the average asymmetry of waves ahead of rogue waves in our dataset was usually close to $\mu = 0.5$, which represents a typical value for regular first-order waves. Furthermore, it can be inferred that the radar devices measured slightly more asymmetric and steep waves than the wave buoys. The tendency of buoys to underestimate wave crests is recognised in the literature (Allender et al., 1989; Forristall, 2000).


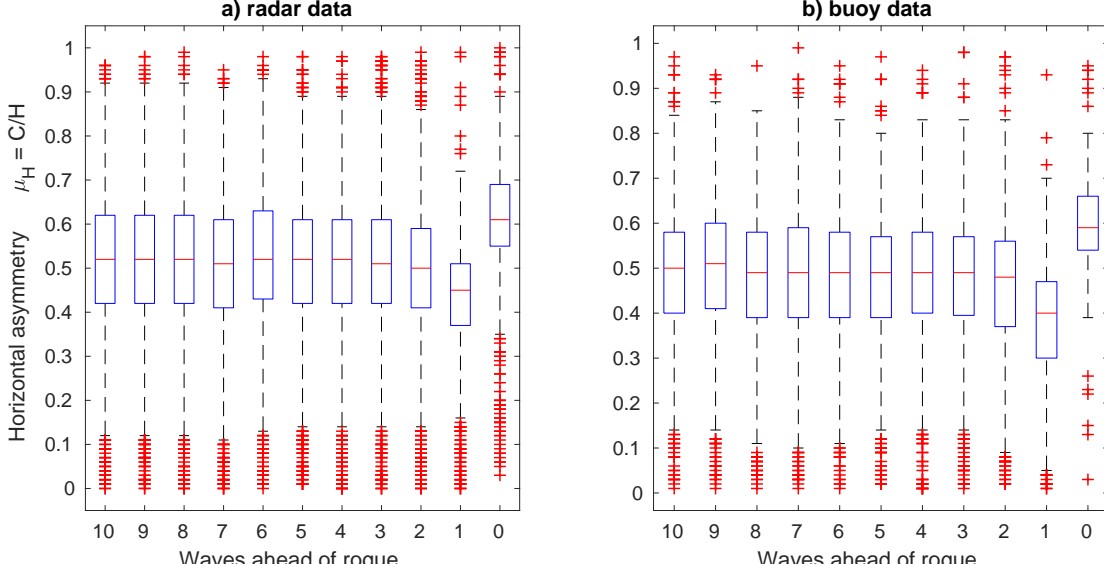

**Figure 14.** Distribution of asymmetry of individual waves ahead of rogue waves (wave 0) for radar (left) and wave buoy (right) locations. Distributions were obtained from 1.234 (716) rogue wave samples at radar (buoy) locations for which the significant wave height exceeded the corresponding long-tern 95% percentile. Distributions are shown as Box-Whisker plots (median: red line; box: inter-quartile range; whiskers: 1.5 × inter-quartile range; red crosses: data outside the whiskers).

## 4   Discussion

Comparison of rogue wave frequencies in our data set revealed that the radar stations usually identified more rogue waves during the measurement period than the wave buoys. Generally, all radar stations were located in the western and all wave buoys in the eastern part of our analysis domain. By means of the available dataset, it is therefore not possible to unambiguously assign

these differences to either the use of different measurement devices or to the location of measurements in different regions. Generally it is known that different wave measurement devices yield different results. Compared to other instruments, wave buoys tend to underestimate the statistics of the amplitude (Allender et al., 1989) and yield statistics below the Gaussian curve (Baschek and Imai, 2011). Possible explanations for these effect were given by Forristall (2000), who concluded that wave





buoys may on the one hand be dragged through or slide away from (short) wave crests, which might result in missing the
maximum amplitudes. On the other hand, these devices tend to cancel the second-order nonlinearities by their own Lagrangian movement and thus overestimate the mean water level, which in turn leads to an underestimation of crest heights (Forristall, 2000). Especially for steep waves, which are strongly nonlinear, this leads to significant differences compared with fixed Eulerian sensors (Longuet-Higgins, 1986). In addition, it must be taken into account that wave buoys are moored and as such represent a part of a damped mechanical system. The influence of the anchoring is not clearly to identify (Forristall, 2000).
Radar systems looking down at the water surface on the other hand may overestimate crests by misinterpreting spray, breaking waves or even fog (Grønlie, 2006). Forristall (2005) noted that there is no standard way for the calibration of measurement instruments and that it is not possible to decide which instrument yields the "most correct" results. Because of these obvious differences that may arise from different sensors we assume, that at least large parts of the observed differences were likely caused by the different measurement techniques used. We can, however, not fully rule out that some differences in rogue wave
frequencies between the different regions do exist. To address this issue, joint installations of wave buoys and radar devices at a location would be desirable.

While we assume that large parts of the observed differences in rogue wave frequencies might be attributed to the use of different sensors, there are some examples in the literature, indicating that rogue wave statistics may differ regionally, for example, due to different fetch, bathymetry or proximity to the coast. Baschek and Imai (2011) found that rogue wave
frequencies were not significantly different in deep and shallow water, but were reduced in sheltered coastal oceans. Cattrell et al. (2018) on the other hand reported that wave frequencies were not spatially uniform and increased in coastal seas. In our case, there was one buoy (SEE) at which more rogue waves than expected from the Forristall distribution were identified. There are several options that may explain this behavior. These options need to be explored further. At first, the buoy is deployed at a rather shallow average water depth. This may lead to measurement issues as described above, in particular in the presence
of breaking waves. Furthermore, the region is characterised by a strongly structured bathymetry with strong gradients and by strong tidal currents, which may both contribute to focusing of wave energy. In fact, SEE reveals very particular bathymetry conditions. Located close to the island of Norderney, the measurement buoy is placed directly above a sudden change in water depth. This stimulates shoaling and refraction leading to an increase in wave height (Goda, 2010). Trulsen et al. (2012) have shown experimentally that the propagation of waves over a slope from deep to shallow water may provoke a maximum in
kurtosis and skewness. Based on their findings, they anticipate a local maximum of rogue wave probability which would be in accordance with observations at SEE, but would need further investigation to be fully confirmed.

We compared the relative wave height distribution in our dataset to the Rayleigh and Forristall distributions. Waseda et al. (2011) found the Forristall distribution to fit well to storm wave records from the northern North Sea, both when regarding his entire dataset of 2.723 records, and when forming subsets along different significant wave heights. Over a range of sea states
and from a large dataset of 122 million waves, Christou and Ewans (2014) found the waves to possess statistical characteristics in between linear and second-order theory. In our data, the distribution of wave heights in the total dataset showed a fair agreement with the Forristall distribution up to a relative wave heights of $HH_s{}^{-1} \gtrsim 2$. Rogue waves, and especially rogue waves with a very large relative wave height occurred more often than expected from the Forristall distribution. Deviations



from this distribution, however, varied across stations and between buoys and radar stations. Magnusson et al. (2003) reported
similar deviations in the upper tail of the relative wave height distribution, although they find the statistics of their analysed
individual wave heights from buoy and laser data in agreement with Rayleigh and Weibull distributions. Forristall (2005)
confirmed an underestimation of large individual wave heights by his distribution when single records were considered, but
could not find such a behavior for larger amounts of data. He concluded that *"a large wave which stands out as unusual in a
short record may be expected if we look long enough. [...] If we wait a long time, Gaussian statistics can produce a very large
wave."* (Forristall, 2005). In fact, Haver and Andersen (2000), who brought up the question whether or not rogue waves can
be considered part of a typical distribution, stated that a statistical approach based on empirical data may not be sufficient to
address this question, as empirical records typically contain too few rogue waves. Even in our large data set, there is only a
small number of 21 cases in which relative wave heights exceeded a factor of $HH_s^{-1} \gtrsim 3$.

## 5 Conclusions

Six years of wave measurements from eleven measurement sites in the southern North Sea were quality controlled and analysed
for rogue wave occurrences and frequency. We found that rogue wave frequencies were relatively constant over seasons and
uncorrelated between stations. We found that on average, the distribution of wave heights followed the Forristall distribution
with some deviations in the upper tail in particular for radar sites. However, deviations are based on estimates from a relatively
small number of cases. While there appeared to be some differences in the wave height distribution in samples with and without
rogue waves, differences were too small for being usable in rogue wave detection. Other properties such as wave steepness or
wave asymmetry did not show substantial differences between samples containing a rogue or not. From the analyses of their
data, Christou and Ewans (2014) suggested that rogue waves may simply represent rare realizations from typical distributions,
caused by dispersive focusing. Using a different dataset, this conclusion is in principle supported by our analyses.

*Author contributions.* All authors contributed to the idea and scope of the paper. IT performed the analyses and wrote the manuscript. RW,
JM and OK provided help with data analysis, discussed the results, and contributed to the writing of the paper. RW supervised the work.

*Competing interests.* The authors declare that they have no conflict of interest.

*Acknowledgements.* This work was supported by the Federal Maritime and Hydrographic Agency (BSH). The buoy data were kindly pro-
vided by BSH and the Lower Saxony Water Management, Coastal Defence and Nature Conservation Agency (NLWKN). The authors are
grateful to G. Feld and Shell for providing the radar data. The authors are grateful to Christian Senet for providing Matlab code for the
visualisation of buoy raw data, and for his valuable input.



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
