# Peer review of "A Statistical Analysis of Rogue Waves in the Southern North Sea"

_Natural Hazards and Earth System Sciences, 2020_

## Referee Comment (RC1) · Anonymous Referee #1 · 22 Apr 2020

**Review**

on the manuscript "A Statistical Analysis of Rogue Waves in the Southern North Sea" by Ina Teutsch, Ralf Weisse, Jens Moeller, and Oliver Krueger, submitted for publication in NHESS.

The paper represents a nice study of a huge dataset of the surface displacement records collected through 6 years by 11 gauges installed in a region of the North Sea. I would say, every study which deals with such a large amount of data is an important event in the oceanography. In my opinion, the manuscript is clearly written with thorough discussion of the results of the processing and with a perfect graphical representation. There is practically no technical drawbacks in the text. I assume, the study is far from exhaustive, and a series of subsequent publications may be anticipated. However, there is a set of important findings and conclusions formulated through the text and in the discussion and conclusions. I congratulate the authors with such a good job done, and believe that the NHESS should be happy to publish this paper. At the same time, I believe that some issues require more clarification / mentioning / discussion, I list them below. They should require a minor modification of the texts which should improve its quality, in particular should help to depict a more general picture of the problem.

1. I suggest another possible explanation of the difference between the statistics obtained by the radar and by the buoys, which is the different sampling frequency. The low frequency of the buoys 1.28Hz may lead to a poor description of short and sharp steep crests. If so, the records by the radars are more trustworthy.

2. The authors compared different subsets of the data (seasonal, in rough sea states, recorded by buoys or radars, etc), but not sorted basing on the spectral properties (presence of one or several wave systems, angle spectrum width, etc.). If the corresponding data is available, can this analysis be performed in the future?

3. The depth conditions should be described and discussed more thoroughly. According to Table 1, the depths seem to correspond to fairly shallow water. In the paper they are divided into shallow and deep with respect to the condition $kh = 1.36$, which is very mild if one wants to observe the effect of the Benjamin – Feir self-modulation. I suggest to indicate the magnitudes of $kh$ (or range of them) in Table 1.

   When comparing with other observations (Sec. 4 Discussion) with respect to the probability distribution laws (Rayleigh or Forristall), the distinction between the depth conditions is not always made. However, the issue whether the PDF is better described by the Rayleigh or the Forristall distribution (and why it differs in different researches) is probably even more important than the peculiarities of the PDF tails.

4. Some seemingly worthy references are absent in the paper; they may be relevant:

   Chien, H., Kao, C.-C., Chuang, L.Z.H. (2002) On the characteristics of observed coastal freak waves. Coast. Eng. J. 44, 301–319.

   Liu, P.C., MacHutchon, K.R. (2006) Are there different kinds of rogue waves? In: Proc 25th Int Conf OMAE 2006, Hamburg, Germany, 2006, OMAE2006-92619:1–6.

   Mori, N., Liu, P.C., Yasuda, T. (2002) Analysis of freak wave measurements in the Sea of Japan. Ocean. Eng. 29, 1399–1414.

   Paprota, M., Przewlocki, J., Sulisz, W., Swerpel, B.E. (2003) Extreme waves and wave events in the Baltic Sea. In: Rogue Waves: Forecast and Impact on Marine Structures. GKSS Research Center, Geesthacht, Germany.

Pinho de, U.F., Liu, P.C., Ribeiro, C.E.P. (2004) Freak waves at Campos Basin, Brazil. Geofizika 21, 53–67.

A review and discussion of the effects of the averaged wave steepness, spectral properties, etc. on the likelihood of the rogue wave occurrence, robustness of the rogue wave estimators may be found in Chapter 1 of the book:

Kharif, C., Pelinovsky, E., Slunyaev, A. (2009) Rogue Waves in the Ocean, Springer-Verlag Berlin Heidelberg.

5. It is mentioned in the end of Sec. 2.1 that the zero-upcrossing method is employed in the study with the reference to Goda (1986) that the down-crossing analysis should lead to the same results. Have the authors checked that the zero-downcrossing approach yields the same PDF? A clear difference between the up and down-crossing analyses was found in the numerical simulations [Sergeeva, A., Slunyaev, A. (2013) Rogue waves, rogue events and extreme wave kinematics in spatio-temporal fields of simulated sea states. Nat. Hazards Earth Syst. Sci. 13, 1759-1771; Slunyaev, A., Sergeeva, A., Didenkulova, I. (2016) Rogue events in spatiotemporal numerical simulations of unidirectional waves in basins of different depth. Natural Hazards 84, 549-565] (rogue wave shapes possessed asymmetry in rough sea states having deeper following troughs), and the in-situ observations [Pinho de, U.F., Liu, P.C., Ribeiro, C.E.P. (2004) Freak waves at Campos Basin, Brazil. Geofizika 21, 53–67]. Can the authors support or deny the difference?

6. End of the caption for Table 2: "*pear year*" should be corrected.

7. To the end of Sec. 2.2. M. Cristou and K. Ewans admitted that the applied quality check procedure rejected as possibly erroneous a much greater portion of time records with rogue waves compared to 'ordinary' time records, what could affect the eventual PDF. Can the authors give the portion of discarded 30-min samples with potential rogue waves versus the portion of discarded records with all waves below the threshold of $2H_s$?

8. Fig. 3: the red and orange lines cannot be distinguished when printed in my color printer. Please change the colors.

9. Fig. 5, 6. Such large anomalies of statistically valid annual rogue wave frequencies may probably mean that the man part of the registered rogue waves was caused by a few localized in time and space sea states which passed through the measurement locations. Are the rogue wave events at a given location clustered in time? This may be a question to be answered in the next paper.

10. Figs. 8, 9 are plotted for large $H_s$. Could you please give some estimation of the threshold value of $H_s$, how large it was.

11. Line 240: please remove the repetition "*at at the 95%*".

12. Lines 285-287. It is not really clear why the two limits of the dispersion relation for shallow and deep water were applied, as most of the conditions seem to correspond to the intermediate depth. What was the interval of $kh$ in the experimental data?

13. Fig. 12. Looking at plots in (a) and (b) ((a) particularly) I would say that in shallow water rogue waves preferably occur in the conditions of a small steepness. Can you agree with this conclusion?

---

## Referee Comment (RC2) · Anonymous Referee #2 · 2 May 2020

**Report on "A statistical analysis of rogue waves in the Southern North Sea" by I. Teutsch, R. Weisse, J. Moeller and O. Krueger**

In this manuscript the authors discuss a dataset of field measurements collected during six years at eleven measurement sites in the southern North Sea. Such a dataset is rare and very valuable. The analysis is presented in an appropriate way and the results are interesting and useful. I recommend publishing this manuscript after the following remarks that may improve the manuscript are considered:

1. In line 17 you refer to the old "highest third" criterion for significant wave height. However, the Rayleigh distribution in equation (1) with the parameter values as specified in line 50 and most of the subsequent developments are probably based on the modern "four standard deviations" criterion, thus you are in fact defining the "significant wave height" in two different ways in your manuscript? Please be explicit about this!

2. In lines 46–50 it is probably not accurate to say that a sea surface that is a stationary Gaussian process has wave heights that are Rayleigh distributed according to equation (1) with the given reference parameters. This approximation is correct for the limit of an infinitely narrow-band process, but needs to be adjusted to account for the effect of finite spectral bandwidth in order not to overpredict the true values, see e.g. Næss (1985).

3. In line 113 you write $\overline{T_z^{-1}}$, but you probably want to write $\overline{T_z}^{-1}$.

4. In equation (2) you should explain that $C$ is crest height.

5. In figures 7 and 9 it is not clear if the red curve conveys information different from the histogram of blue bars? If you declare the total number of waves and the bin width (which you should declare) then these two graphical representations are equivalent?

6. In lines 280–281 you use the water depth $kh = 1.36$ "below which nonlinear instabilities are mostly absent (Benjamin & Feir, 1967)". It is true that if you limit to uniform waves with long-crested perturbations you find instability only for $kh > 1.36$ (Benjamin & Feir, 1967). If you allow the uniform waves to have short-crested perturbations you also find instability below this threshold (Benney & Roskes, 1969). For your field measurements you have to accept that your waves are short-crested.

7. In lines 280–281 you use the water depth $kh = 1.36$ to "distinguish between deep and shallow water waves", in order to select between two different asymptotic approximations of the full dispersion relation. However, this is a stability limit for long-crested waves, and does not distinguish different asymptotic shapes of the dispersion relation.

   Please explain why you want to approximate the full dispersion relation here rather than to use the full relation!

8. In line 373 you refer to Trulsen *et al.* (2012) for experimental results regarding kurtosis and skewness over a bottom slope. You may also want to refer to Trulsen *et al.* (2020) where substantially more experimental evidence was presented for kurtosis and skewness over a shoal. From their work there appears to be different depth regimes with different behaviors. It could be interesting if you discuss your depths and observations in relation to the different depth regimes anticipated by Trulsen *et al.* (2020).

**References**

Benjamin, T. B. & Feir, J. E. 1967 The disintegration of wave trains on deep water. *J. Fluid Mech.* **27**, 417–430.

Benney, D. J. & Roskes, G. J. 1969 Wave instabilities. *Studies Appl. Math* **48**, 377–385.

Næss, A. 1985 The joint crossing frequency of stochastic-processes and its application to wave theory. *Appl. Ocean Res.* **7**, 35–50.

Trulsen, K., Raustøl, A., Jorde, S. & Rye, L. B. 2020 Extreme wave statistics of longcrested irregular waves over a shoal. *J. Fluid Mech.* **882**, R2.

Trulsen, K., Zeng, H. & Gramstad, O. 2012 Laboratory evidence of freak waves provoked by non-uniform bathymetry. *Phys. Fluids* **24**, 097101.

---

## Author Comment (AC1) · 22 Jun 2020

We thank Referee #1 for the constructive comments that have helped us to clarify and improve some points in our manuscript. In the following, we show how we will address the individual issues raised by the reviewer in the revised manuscript.

1. We thank the reviewer for this comment, which we believe addresses an important point. The comment suggests that the different statistics may also be the result of different sampling frequencies of buoys and radars. To address this point we will subsample statistics from a radar time series with a frequency of 1 Hz (close to the buoy frequency), compare them with statistics obtained from the entire time series and we will add a corresponding discussion in the manuscript.

[Figure]

2. We agree with the reviewer that the information on spectral properties available from buoy measurements and hindcast data represent important and interesting aspects. We think, however, that this would be beyond the scope of the present manuscript. As suggested by the reviewer we would leave this for investigation in a follow-up paper.

3. We thank the reviewer for raising this important issue. We agree and will include typical kh ranges in Table 1. We will also include depth conditions in the discussion when comparing our data with results from previous studies.

4. We thank the reviewer for suggesting further references and will consider and include them in the revised manuscript.

5. We acknowledge that different literature provides different results and conclusions. We did not explicitly test whether the different approaches lead to different results for our data but chose one of the definitions. We will, however, include a corresponding discussion of the different views and results in the revised manuscript.

6. The typing error will be corrected in the revised manuscript.

7. We understand the point the reviewer makes. We checked the rejected samples and found that providing such a ratio would be difficult as most of the rejected samples were related to obvious transmission problems where individual measurements or shorter periods contained unreasonable data and most samples therefore could not be associated with one or the other group. We, therefore, decided not to present such a ratio.

8. Colors will be changed accordingly in the revised version.

9. Thank you very much for this idea which, as suggested, is worth looking at in a following manuscript.

10. To address this point, in the revised version we will add the values of the 95th percentile of significant wave height at each station in a table supporting the Figures.

11. The typing error will be corrected.

12. To address this point we will provide typical ranges for kh in the revised version (see reply to comment # 3).

13. The reviewer is correct. Following the suggestion of Referee #2, we will repeat the analysis using the full dispersion relation and then comment on the identified tendency.
* * *

---

## Author Comment (AC2) · 22 Jun 2020

We thank Referee #2 for the constructive comments that have helped us to clarify and improve some points in our manuscript. In the following, we show how we will address the individual issues raised by the reviewer in the revised manuscript.

1. We thank the reviewer for highlighting the different definitions of the significant wave height. However, we do not fully agree here. We used the traditional mean of the "highest third" criterion for $H_s$ throughout the study. The "four standard deviations" criterion can be obtained from the Rayleigh distribution, which nevertheless does not affect our choices and parameters (cf. e.g. Holthuisen 2007, p. 68ff). Our approach is consistent with approaches taken in other

observational-based studies [Soares (2003), Magnusson (2003), Stansell (2004), Waseda (2011), Baschek and Imai (2011)].

2. We are grateful to the reviewer for pointing out that our assumptions are open to debate to some extent. We completely agree with the reviewer. We will revise the text accordingly taking the narrow-band assumption and the cited reference into account.

3. The typing error will be corrected.

4. We will introduce the definition in the text.

5. In Figures 7 and 9, the red curve shows the frequency with which a given relative wave height was exceeded in the measurement data, without information on how many waves this result is based on. If, for example, the red curve shows that approx. 1 in 6,000 waves is a rogue wave, we could not immediately infer how many waves this result is based on. Therefore, the blue bars also show the absolute number of waves measured in each category. The number can be read from the y-axis on the right-hand side of the figure. We have divided the relative wave height axis into 100 bins. We present the blue bars additionally to the curves, to clarify that the findings on the right edge of the figure are based only on a few measured waves. We will rewrite the caption to make this more explicit and clearer.

6. The reviewer is right and we will modify the text accordingly. Also following the suggestion by Referee #1 we provide information on typical kh values for each site.

7. We are grateful for this comment, which is relevant for our analyses shown in Figure 12. The comment led us to rethink our choice of how we approximated the dispersion relation. We will now use the full dispersion relation in the revised manuscript.
8. Thank you very much for the additional reference and the suggested interesting analysis. We will include this point into the discussion in a revised manuscript but we will leave the full analysis to a follow-up manuscript.

**References:**

Baschek, B. and Imai, J.: Rogue Wave Observations Off the US West Coast, Oceanography, 24, 158–165, https://doi.org/10.5670/oceanog.2011.35, https://doi.org/10.5670/oceanog.2011.35, 2011.

Holthuijsen, L. H.: Waves in Oceanic and Coastal Waters, Cambridge University Press, https://doi.org/10.1017/cbo9780511618536, https://doi.org/10.1017/cbo9780511618536, 2007.

Magnusson, A. K., Jenkins, A., Niedermayer, A., and Nieto-Borge, J. C.: Extreme Wave Statistics from Time-Series Data, in: Proceedings 465 of MAXWAVE Final Meeting, Geneva, pp. 17: 231–245, 2003.

Soares, C. G., Cherneva, Z., and Antão, E.: Characteristics of abnormal waves in North Sea storm sea states, Applied Ocean Research, 25, 337–344, https://doi.org/10.1016/j.apor.2004.02.005, https://doi.org/10.1016/j.apor.2004.02.005, 2003.

Stansell, P.: Distributions of freak wave heights measured in the North Sea, Applied Ocean Research, 26, 35–48, 475 https://doi.org/10.1016/j.apor.2004.01.004, https://doi.org/10.1016/j.apor.2004.01.004, 2004.

Waseda, T., Hallerstig, M., Ozaki, K., and Tomita, H.: Enhanced freak wave occurrence

with narrow directional spectrum in the North Sea, Geophysical Research Letters, 38, https://doi.org/10.1029/2011gl047779, https://doi.org/10.1029/2011gl047779, 2011.

---

## Author Response (AR1)

We thank both reviewers for their constructive comments that have helped us to clarify and improve some points in our manuscript. In the following, we show how we addressed the individual issues raised by the reviewers in the revised manuscript.

**Reviewer #1**

1. We thank the reviewer for this comment, which we believe addresses an important point. The comment suggests that the different statistics may also be the result of different sampling frequencies of buoys and radars. To address this point we subsampled statistics from the radar time series with a frequency of 1 Hz (close to the buoy frequency) and compared them with statistics obtained from the entire time series. A corresponding discussion was added to the manuscript (lines 375-381 in the marked-up manuscript version).

2. We agree with the reviewer that the information on spectral properties available from buoy measurements and hindcast data represent important and interesting aspects. We think, however, that this would be beyond the scope of the present manuscript. As suggested by the reviewer we would leave this for investigation in a follow-up paper.

3. We thank the reviewer for raising this important issue. We agree and included typical kh ranges in Table 1. We also included depth conditions in the discussion when comparing our data with results from previous studies (lines 403-408, 418-420 in the marked-up manuscript version).

4. We thank the reviewer for suggesting further references. We have included references to the papers by Chien et al. [2002] (lines 29, 68), Mori et al. [2002] (lines 29, 68), Paprota et al. [2003] (lines 304, 310), Pinho et al. [2004] (lines 66, 416), and the book by Kharif et al. [2009] (lines 42-44). We decided not to include a reference to the paper by Liu et al. [2006], as in our data we did not identify rogue waves exceeding H/Hs=4.

5. We acknowledge that different literature provides different results and conclusions. We did not explicitly test whether the different approaches lead to different results for our data but chose one of the definitions. We have, however, included a corresponding discussion of the different views and results (lines 411-417 in the marked-up manuscript version).

6. The typing error was corrected in the revised manuscript.

7. We understand the point the reviewer makes. We checked the rejected samples and found that providing such a ratio would be difficult as most of the rejected samples were related to obvious transmission problems where individual measurements or shorter periods contained unreasonable data and most samples therefore could not be associated with one or the other group. We, therefore, decided not to present such a ratio.

8. Colors were changed in the revised version.

9. Thank you very much for this idea which, as suggested, is worth looking at in a following manuscript.

10. To address this point, we added the values of the 95[th] percentile of significant wave height at each station in a new table (Table 4) supporting the Figures.

11. The typing error was corrected.

12. To address this point we provided typical ranges for kh in the revised version (see reply to comment # 3).

13. The reviewer is correct. Following the suggestion of Referee #2, we repeated the analysis using the full dispersion relation, replaced Figure 12 with the corresponding update, and modified the discussion accordingly (lines 291-312 in the marked-up manuscript version).

**Reviewer #2**

1. We thank the reviewer for highlighting the different definitions of the significant wave height. We used the traditional *mean of the highest third criterion* for $H_s$ throughout the study. This approach is consistent with approaches taken in other observational-based studies [e.g., Soares et al. 2003; Magnusson et al. 2003; Stansell et al. 2004; Waseda et al. 2011; Baschek and Imai 2011]. To address the issue, in the revised version we added a note on the different definitions (lines 56-58 in the marked-up manuscript version).

2. We are grateful to the reviewer for pointing out that our assumptions made are open to debate to some extent. We completely agree with the reviewer. We revised the text accordingly taking the narrow-band assumption and the cited reference into account (lines 49, 59-62 in the marked-up manuscript version).

3. The typing error was corrected.

4. We introduced the definition in the text (line 139 in the marked-up manuscript version).

5. To make this clearer, we have rewritten the Figure captures and added a line of explanation to the text (lines 201-202 in the marked-up manuscript version).

6. The reviewer is right and we modified the text accordingly (lines 290ff in the marked-up manuscript version). Also following the suggestion by reviewer #1 we provide information on typical kh values for each site (Table 1).

7. We are grateful for this comment, which is relevant for our analyses shown in Figure 12. The comment led us to rethink our choice of how we approximated the dispersion relation. We repeated the analysis using the full dispersion relation, replaced Figure 12 with the corresponding update, and modified the discussion accordingly (lines 291-312 in the marked-up manuscript version).

8. Thank you very much for the additional reference and the suggested interesting analysis. We included this point and the reference into the discussion (lines 399-400 in the marked-up manuscript version), but we will leave the full analysis to a follow-up manuscript.

**References**

See the list of references in the revised manuscript.

[revised manuscript text omitted]

**List of changes**